# Pentraxin-3 Modulates Osteogenic/Odontogenic Differentiation and Migration of Human Dental Pulp Stem Cells

**DOI:** 10.3390/ijms20225778

**Published:** 2019-11-17

**Authors:** Yeon Kim, Joo-Yeon Park, Hyun-Joo Park, Mi-Kyoung Kim, Yong-Il Kim, Hyung Joon Kim, Soo-Kyung Bae, Moon-Kyoung Bae

**Affiliations:** 1Department of Oral Physiology, Dental and Life Science Institute, School of Dentistry, Pusan National University, Yangsan 50610, Korea; graceyeon88@gmail.com (Y.K.); pmh2879@naver.com (J.-Y.P.); phj3421@hanmail.net (H.-J.P.); eenga@naver.com (M.-K.K.); hjoonkim@pusan.ac.kr (H.J.K.); 2Department of Orthodontics, School of Dentistry, Pusan National University, Yangsan 50610, Korea; kimyongil@pusan.ac.kr; 3Department of Dental Pharmacology, School of Dentistry, Pusan National University, Yangsan 50610, Korea

**Keywords:** pentraxin-3, human dental pulp stem cells, osteogenic/odontogenic differentiation

## Abstract

Pentraxin-3 (PTX3) is recognized as a modulator of inflammation and a mediator of tissue repair. In this study, we characterized the role of PTX3 on some biological functions of human dental pulp stem cells (HDPSCs). The expression level of PTX3 significantly increased during osteogenic/odontogenic differentiation of HDPSCs, whereas the knockdown of PTX3 decreased this differentiation. Silencing of PTX3 in HDPSCs inhibited their migration and C-X-C chemokine receptor type 4 (CXCR4) expression. Our present study indicates that PTX3 is involved in osteogenic/odontogenic differentiation and migration of HDPSCs, and may contribute to the therapeutic potential of HDPSCs for regeneration and repair.

## 1. Introduction

Until recently, inflammation in dental pulp was considered to exert only detrimental effects, but there is growing evidence regarding the consideration of the inflammatory process as a prerequisite for healing and regeneration in the dentin–pulp complex [1]. Human dental pulp stem cells (HDPSCs), which are present within the “cell-rich zone” of the dental pulp, are a source of cells to replace damaged cells at a site of injury [2]. HDPSCs locally interact with the inflammatory microenvironment, which probably affects their fate, leading to their proliferation, migration, and differentiation into odontoblast lineage cells for the restoration of dental pulp and dentin [3].

Pentraxins are a superfamily of multimeric proteins, characterized by the conserved C-terminal pentraxin domain [4]. Depending on their primary structure, pentraxins are classified into short and long pentraxins [5]. C-reactive protein (CRP) and serum amyloid P (SAP) are the short pentraxins, whereas pentraxin-3 (PTX3) is the prototypic member of the long pentraxins [5]. CRP and SAP are mainly expressed in hepatocytes in response to interleukin-6 (IL-6), whereas PTX3 is produced by different cell types at sites of local infection and inflammation [6]. Growing evidence suggests that PTX3 exerts protective effects in addition to its negative impacts in experimental models of inflammation [7]. We previously reported that PTX3 was inducible in human dental pulp cells upon stimulation with tumor necrosis factor-α (TNF-α) [8]; however, to our knowledge, the function of PTX3 in HDPSCs has not been reported until now. Therefore, the aim of the present study was to investigate the role of PTX3 in the proliferation, migration, and osteo/odontogenic differentiation of HDPSCs, and to elucidate the underlying mechanisms.

## 2. Results

### 2.1. Expression of PTX3 during Osteo/Odontogenic Differentiation of HDPSCs

To study the expression of PTX3 during osteo/odontogenic differentiation, HDPSCs were cultured in osteogenic medium for 14 days and their differentiation status was evaluated using alkaline phosphatase (ALP) staining (Figure 1A). mRNA expression levels of ALP and dentin matrix protein-1 (DMP-1), the osteo/odontogenic differentiation markers, were estimated by real-time PCR analysis (Figure 1B). As shown in Figure 1B, mRNA expression of PTX3 significantly increased during osteo/odontogenic differentiation of HDPSCs; this upregulation was also reflected in the expression levels of PTX3 protein, as revealed by Western blot analyses (Figure 1C), and the amount of secreted PTX3 protein, as seen with ELISA (Figure 1D).

### 2.2. Inhibition of PTX3 Impairs Osteo/Odontogenic Differentiation of HDPSCs

In HDPSCs transduced with lentivirus carrying PTX3-specific shRNA, protein levels of PTX3 were significantly lower than in those transduced with non-specific shRNA. This confirmed that in HDPSCs, PTX3 was effectively knocked down by specific shRNA (Figure 2A). To explore the role of PTX3 in osteo/odontogenic differentiation of HDPSCs, virus-transduced HDPSCs were cultured in mineralization medium for 2 weeks. A significantly lower level of ALP staining was observed in the PTX3 shRNA-transduced HDPSCs at each time point of differentiation compared to control shRNA-transduced HDPSCs (Figure 2B). Real-time PCR analysis showed that mRNA levels of the identified osteo/odontogenic markers, DMP-1 and dentin sialophosphoprotein (DSPP), were significantly lower in PTX3 shRNA-transduced HDPSCs than in the control group at 14 days of differentiation (Figure 2C).

### 2.3. Knockdown of PTX3 Inhibits Migration of HDPSCs

The effect of PTX3 on the proliferation of HDPSCs was assessed using MTT assay. As shown in Figure 3A, knockdown of PTX3 had no effect on the proliferation of HDPSCs. Next, we carried out a scratch wound migration assay to observe the motility of PTX3 silencing in HDPSCs. PTX3-knockdown HDPSCs showed a reduction in the number of cells that migrated into the scratch wound compared with that observed in control shRNA-transduced HDPSCs (Figure 3B). We evaluated the function of PTX3 in relation to the migration of HDPSCs using Boyden’s chamber migration assay. PTX3 silencing significantly inhibited the chemotactic movement of HDPSCs compared with HDPSCs transduced with control shRNA (Figure 3C).

### 2.4. Knockdown of PTX3 Regulates SDF-1/CXCR4 Axis in HDPSCs

C-X-C chemokine receptor type 4 (CXCR4), a G-protein coupled receptor, regulates the migration of HDPSCs to the sites of injury through binding to SDF-1 (stromal cell derived factor-1), which plays a critical role in dental pulp inflammation and regeneration [9,10,11]. Silencing of PTX3 in HDPSCs significantly decreased the expression of CXCR4 mRNA, as measured by real-time RT-PCR (Figure 4A, left), leading to down-regulation of the CXCR4 protein, as seen by Western blotting (Figure 4A, right). A previous report showed that SDF-1 could induce the expression of CXCR4 in HDPSCs [12]. We found that SDF-1-induced CXCR4 expression was significantly inhibited in HDPSCs transduced with PTX3 shRNA compared to controls (Figure 4B). Furthermore, PTX3 silencing with PTX3 shRNA in HDPSCs significantly impaired their migration toward SDF-1 in the transwell migration assays (Figure 4C).

## 3. Discussion

*PTX3*, the first gene from the long-pentraxin family to be cloned, encodes an acute-phase protein that plays a pleiotropic role in various inflammatory diseases [13]. Unlike short pentraxins (e.g., CRP and SAP), which are induced by IL-6 in the liver, PTX3 is induced by proinflammatory cytokines, microbial moieties, and agonists of toll-like receptor (TLR) in different types of cells [14]. Growing evidence demonstrates the deleterious as well as the protective roles of PTX3 in different inflammatory models [7,15]. PTX3 promotes the development of lung injury via upregulation of pro-inflammatory molecules [16], and induces endothelial cell dysfunction by evoking morphological alterations in them [17]. Nonetheless, blocking PTX3 action attenuates tissue inflammation and cytokine production after intestinal ischemia and reperfusion injury [18]. PTX3 also confers resistance to endotoxic shock induced by lipopolysaccharides (LPS) [19] and supports brain repair after ischemic injury by glial scar formation and resolution of edema [20]. PTX3 acts as a negative feedback regulator that controls inflammation by dampening excessive leukocyte recruitment [21]. Recent studies highlighted that PTX3 participated in the resolution of inflammation [22], which is now recognized as an active process that reduces leukocytes, clears debris from inflamed or injury sites, and recruits resident stem/progenitor cells to restore tissue homeostasis [23]. Until recently, inflammation was thought to have undesirable effects on dental pulp [24], but growing evidence indicates that low-grade inflammation is crucial for repair and regeneration in the dentin–pulp complex [25]. Further investigations are needed to explore whether PTX3 is possibly a homeostatic regulator in the interplay between inflammation and regeneration in dental pulp.

Untreated injuries of dental pulp, caused by trauma or infection, lead to chronic inflammatory reaction and pulp necrosis, but if the harmful irritation is alleviated before the progression of tissue necrosis, healing of the pulp tissue is possible [24]. Recently, HDPSCs isolated from inflamed dental pulp tissue were shown to retain some of their pluripotent characteristics and regeneration potential [26]. Low levels of pro-inflammatory mediators induced the differentiation of HDPSCs to odontoblast-like cells, leading to dentine regeneration [1]. PTX3 is probably involved in pathogenesis, and its levels correlate with disease severity [15,27]. Recently, PTX3 was suggested as a candidate biomarker to reflect and monitor inflammation status [28]. Plasma levels of PTX3 may be a useful indicator for the assessment of vascular inflammation and neointimal thickening [29]. Expression levels of PTX3 in prostatic tissue and its serum levels predict the progression of prostate inflammation to prostate cancer [30]. A high concentration of PTX3 in gingival crevicular fluid (GCF), saliva, and plasma correlates with the severity of periodontal disease, as defined by clinical evaluation parameters [31,32]. In our previous study, we demonstrated the function of PTX3 in the regulation of pulp inflammation induced by TNF-α [8]. Hence, diagnostic or prognostic criteria could be clinically correlated with PTX3 expression in the pathogenesis of pulpal inflammation, which could lead to the identification of new markers for pulp inflammation.

Dental pulp has the ability to regenerate and repair dentin in response to dental injuries caused by trauma or infection, which is known as reparative dentinogenesis. The formation of reparative dentin results from the recruitment, proliferation, and differentiation of HDPSCs [33]. After tooth injury, HDPSCs localized in a perivascular area were attracted to damaged sites and differentiated into odontoblast-like cells, which produced the reparative dentin [34]. SDF-1 encouraged tissue healing and regeneration by recruiting CXCR4-positive stem cells/ progenitors to the damaged area [35]. Inflamed human dental pulp contained high levels of SDF-1 expression, which was positively detected in infiltrative inflammatory cells and endothelial cells of the microvessels near the carious regions, rather than in normal pulp [10,36]. SDF-1 expression was upregulated by odontogenic induction, inflammatory stimuli, and hypoxic signals in dental pulp cells [9,36,37]. Therefore, the inflammation triggered by dental injury might produce various stimuli, such as proinflammatory molecules and hypoxic signals, thereby inducing the upregulation of SDF-1 [36]. Other studies hypothesized that dental pulp cells at the injured site were stimulated to express and secrete SDF-1, thereby activating CXCR4-positive HDPSCs and subsequently recruiting them toward the damaged site based on the SDF-1 gradient for reparative dentin formation [36,38]. We observed in this study that knockdown of PTX3 impaired the expression of CXCR4 and the cell migration ability of HDPSCs, indicating that PTX3 could affect the migration of HDPSCs toward SDF-1. Based on these observations, it is possible that the existence of CXCR4-positive HDPSCs with intact PTX3 and application of SDF-1 can increase the efficiency of biologically-based therapeutic approaches to regenerate or repair the dental pulp complex.

In conclusion, the present findings demonstrate the significant role of PTX3 as a modulator of migration and osteogenic/odontogenic differentiation of HDPSCs. This study may provide valuable clues regarding the development of HDPSC-mediated regenerative therapies, including dental pulp regeneration and repair.

## 4. Materials and Methods

### 4.1. Reagents

A human PTX3 Quantikine ELISA kit was purchased from R&D Systems (Minneapolis, MN, USA). Human anti-PTX3 antibody and CXCR4 antibody were obtained from Abcam (Cambridge, UK) and Thermo Fisher Scientific (Rockford, IL, USA), respectively. Human α-tubulin antibody was acquired from Biogenex (Fremont, CA, USA). Recombinant SDF-1α was purchased from PeproTech (Rocky Hill, NJ, USA). Methyl-thiazolyl-tetrazolium (MTT), β-glycerophosphate disodium salt pentahydrate, ascorbic acid, dexamethasone, and the alkaline phosphatase kit for ALP staining were all purchased from Sigma-Aldrich (St. Louis, MO, USA).

### 4.2. Cell Culture

The HDPSCs was purchased from Lonza (PT-5025) and maintained in α-modification of Eagle’s minimum essential medium (α-MEM)(Gibco BRL, Grand Island, NY, USA), supplemented with 10% fetal bovine serum (Merck Millipore, Burlington, MA, USA), 1% penicillin–streptomycin (Gibco BRL, USA), and 5 µg/mL Plasmocin (InvivoGen, San Diego, CA, USA), at 37 °C in a humidified incubator with 5% CO_2_.

### 4.3. shRNA Lentiviral Particle Transduction

The negative control shRNA and PTX3shRNA lentiviral particles were obtained from Santa Cruz Biotechnology (Dallas, TX, USA). HDPSCs (1 × 10^6^) were seeded in a T75 flask. Upon reaching 100% confluence, the culture medium was replaced with fresh medium containing 4 µg/mL of polybrene, followed by transduction with control shRNA or PTX3shRNA lentiviral particles. After a change of culture medium, cells were incubated overnight before selection by treatment with 5 µg/mL of puromycin for 5 days. Successful knock down was confirmed with real-time PCR analyses and Western blotting.

### 4.4. Cell Proliferation Assay

The HDPSCs were seeded in 24-well plates, and incubated at 37 °C for 24, 48, or 72 h. At the end of the culture period, cells were placed in fresh medium containing 0.5 mg/mL MTT solution and incubated for 4 h before the addition of 200 µL dimethyl sulfoxide (DMSO). The resultant blue formazan product was measured using a microplate reader at a wavelength of 540 nm.

### 4.5. Alkaline Phosphatase (ALP) Staining

HDPSCs were seeded in 48-well culture plates at a density of 5 × 10^4^ cells/well and cultured in osteogenic differentiation medium, containing 10 mM β-glycerophosphate, 50 µg/mL ascorbic acid, and 0.1 mM dexamethasone for 3, 7, and 14 days, refreshing the medium every 2 days. Using an ALP staining kit (86R-1KT, Sigma-Aldrich), ALP activity was estimated and ALP-positive areas were subjected to quantitative analysis. For this, an alkaline-dye mixture was prepared by mixing 100 µL each of FRV-alkaline solution and sodium nitrate solution and incubating for 2 min at room temperature, followed by diluting with 4.7 mL of distilled water and supplementing with 100 µL of naphthol AS-BI alkaline solution. HDPSCs were fixed with ALP fixing solution for 30 s at room temperature. Next, the cells were washed twice with distilled water before the addition of 200 µL alkaline-dye mixture to each well and incubated for 10 to 25 min in the dark.

### 4.6. Real-Time PCR Analysis

Total RNA was isolated from HDPSCs with a RiboEx reagent kit (Geneg All, Seoul, Korea). cDNA synthesis was carried out using a reverse transcription kit (Promega, Madison, WI, USA) on 2 µg of total RNA. Real-time PCR quantification was performed using a SYBR Green premix (Applied Biosystems, Foster City, CA, USA).The oligonucleotide primers for real-time PCR were designed as follows: β-actin, 5′-ACTCTTCCAGCCTTCCTTCC-3′ and 5′-TGTTGGCGTACAGGTCTTTG-3′; PTX3, 5′-AATGCATCTCCTTGCGATTC3′ and 5′-TGAAGAGCTTGTCCCATTCC-3′; ALP, 5′-ATTTCTCTTGGGCAGGCAGAGAGT-3′ and 5′-ATCCAGAATGTTCCACGGAGGCTT-3′; DMP-1, 5′-AGGAAGTCTCGCATCTCAGAG-3 and 5′-TGGAGTTGCTGTTTTCTGTAGAG-3′; DSPP, 5′-CTGTTGGGAAGAGCCAAGATAAG-3′ and 5′-CCAAGATCATTCCATGTTGTCCT-3′. Amplification conditions consisted of 1 cycle at 95 °C for 10 min, followed by amplification for 40 cycles, each at 95 °C for 15 s, 60 °C for 60 s, and 72 °C for 7 s. Subsequently, a melting curve program was applied with continuous fluorescence measurement. The entire cycling process, including data analysis, took less than 1 h and was monitored using Applied Biosystems.

### 4.7. Western Immunoblot Analysis

The proteins were separated by SDS-PAGE and transferred to a nitrocellulose membrane (GE Healthcare Life Sciences, Marlborough, MA, USA). The membrane was blocked with 5% skim milk in Tris-buffered saline (TBS) containing 0.1% Tween-20 for 1 h at room temperature and probed with appropriate antibodies. A signal was developed with the enhanced chemiluminescence solution (ECL) (GE Healthcare Life Sciences) and was monitored using the LAS4000 (GE Healthcare Life Sciences).

### 4.8. Enzyme-Linked Immunosorbent Assays (ELISA)

HDPSCs were cultured in osteogenic differentiation medium for 0, 3, 7, and 14 days. Subsequently, the culture media were collected and centrifuged at 14,000 rpm for 5 min. The supernatants were stored in aliquots at −70°C. The amount of PTX3 secreted into the medium was determined using the human PTX3 Quantikine ELISA Kit (R&D Systems).

### 4.9. Transwell Chemotactic Migration Assay

SDF-1α (50 ng/mL) in α-MEM with 10% FBS was placed in the lower chambers of the transwells containing 8-mm membrane inserts (Corning Life Sciences, Tewksbury, MA, USA). Transduced HDPSCs (1 × 10^5^), either with control shRNA or PTX3 shRNA, were added to the upper chamber of each transwell and allowed to migrate for 24 h. HDPSCs that migrated through the filter and appeared on the lower side were fixed by careful immersion of the filter in methanol (SK Chemicals, Sungnam, Korea) for 1 min, stained with hematoxylin–eosin solution (DAKO, Hamburg, Germany), rinsed with Milli-Q water (Millipore), and counted in 3 random fields per well. The number of cells in the lower chamber was then counted using a Nikon (Tokyo, Japan) digital sight DS-SMc camera attached to a Nikon ECLIPSE 55i microscope. Each experiment was performed in duplicate, and 3 separate experiments were performed in each group.

### 4.10. Scratch Wound Migration Assay

HDPSCs (5 × 10^5^ cells/well in α-MEM with 10% FBS), were seeded in 60 mm plates and incubated until an adherent confluent monolayer formed. A 10 µL pipette tip was used to create a scratch in the monolayer, and the cells were washed twice with the culture medium. After a 24 h incubation, images of the scratch wounds were captured, and the number of cells in the scratch area was counted using a Nikon digital sight DS-SMc camera attached to a Nikon ECLIPSE 55i microscope. Each experimental group was compared with its respective control. The experiments were repeated three times.

### 4.11. Statistical Analysis

Data represent the mean standard deviation obtained for at least 3 independent experiments. Statistical comparisons between groups were conducted using one-way analysis of variance followed by Student’s *t*-test.

## Figures and Tables

**Figure 1 ijms-20-05778-f001:**
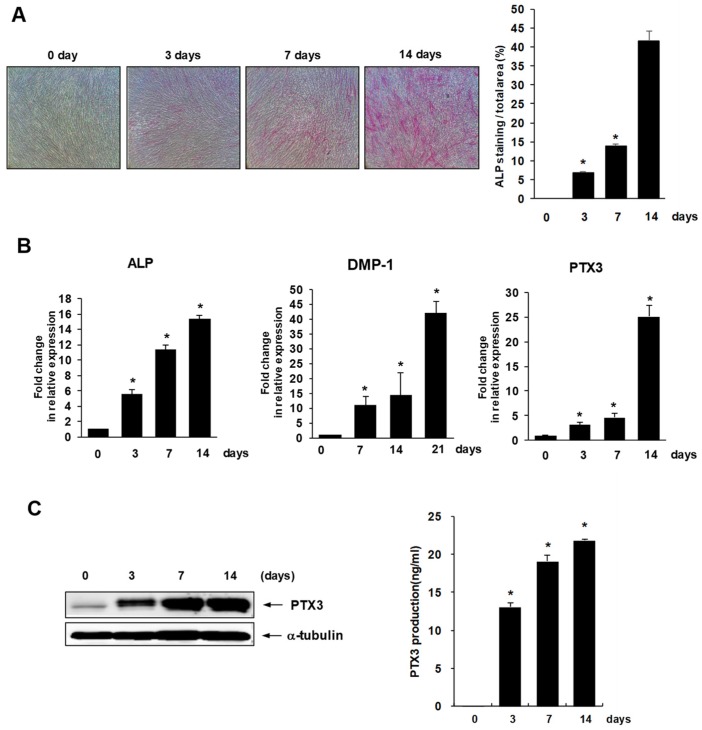
Expression of Pentraxin-3 (PTX3) during osteo/odontogenic differentiation in human dental pulp stem cells (HDPSCs). (**A**) HDPSCs were cultured in osteogenic medium for the indicated number of days. Alkaline phosphatase (ALP) staining was carried out on days 3, 7, and 14. Stained cells were observed by a phase contrast microscope at 100× magnification. ALP-positive areas of HDPSCs were measured by densitometry in triplicate experiments. * *p* < 0.05 compared to day 0. (**B**) Total RNA was isolated during osteogenic differentiation and analyzed by real-time RT-PCR. The expression levels of human *ALP*, *DMP-1*, and *PTX3* mRNA were quantified. The expression levels in the untreated control were considered to be 1.0, and the values were normalized to β-actin mRNA levels. * *p* < 0.05 compared to day 0. (**C**) After culturing HDPSCs in osteogenic media for the indicated number of days, human PTX3 protein levels were determined by Western blot analysis. α-tubulin served as the loading control (left). (**D**) After osteogenic induction of HDPSCs, secreted PTX3 proteins in the supernatants were determined with ELISA (O.D. at 450 nm). * *p* < 0.001 compared to day 0.

**Figure 2 ijms-20-05778-f002:**
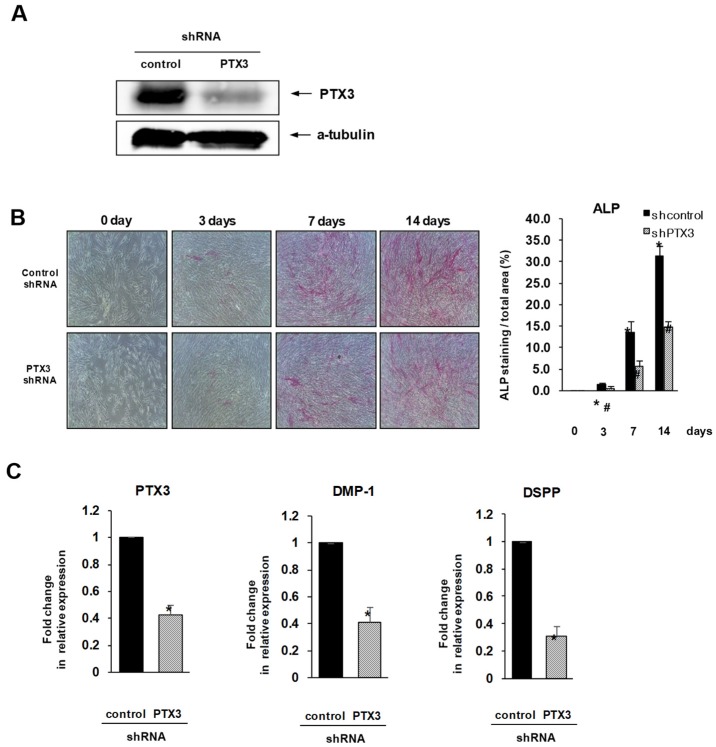
Effect of PTX3 knockdown on the expression of osteo/odontogenic differentiation markers in human dental pulp stem cells. (**A**) HDPSCs were stably transduced with lentiviruses encoding PTX3 shRNA or control shRNA. PTX3 expression was analyzed using Western blot analysis. (**B**) HDPSCs transduced with PTX3 shRNA or the negative control shRNA were seeded in osteogenic media for the indicated number of days. ALP staining was conducted at days 3, 7, and 14. Stained cells were observed by a phase contrast microscope at 100× magnification. ALP-positive areas of HDPSCs were measured by densitometry in triplicate. # *p* < 0.02 compared to control shRNA-transduced HDPSCs. (**C**) HDPSCs, transduced with either PTX3 shRNA or the negative control shRNA, were cultured in osteogenic media for 14 days. The expression levels of *DMP-1*, *DSPP*, and *PTX3* mRNA were evaluated using real-time PCR. The expression level of the control (control shRNA-transduced HDPSCs) was set as 1.0, and the values were normalized to β-actin mRNA levels. * *p* < 0.01 compared to control shRNA-transduced HDPSCs.

**Figure 3 ijms-20-05778-f003:**
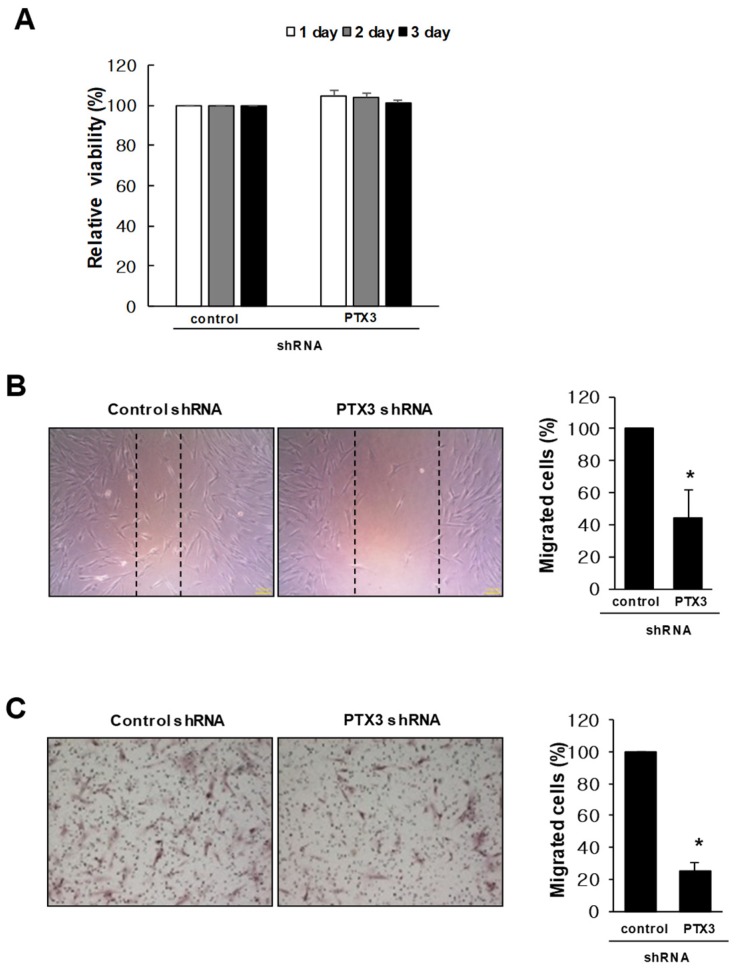
Effect of PTX3 knockdown on migration in human dental pulp stem cells. (**A**) The proliferation of HDPSCs, which were stably transduced with either PTX3 shRNA or the negative control shRNA, was determined by MTT assay. (**B**) Scratch wound migration assay was performed using HDPSCs that were stably transduced with either PTX3 shRNA or the negative control shRNA. After incubation for 24 h, migrated cells were observed with a phase contrast microscope at 100× magnification. The number of cells that migrated beyond the reference line was counted. Each result represents the mean value from triplicate experiments in each group. (**C**) Control shRNA-transduced or PTX3 shRNA-transduced HDPSCs were incubated in transwell upper chambers for 24 h. The migrated cells were stained with hematoxylin–eosin, photographed, and counted. * *p* < 0.05 compared to control shRNA-transduced HDPSCs.

**Figure 4 ijms-20-05778-f004:**
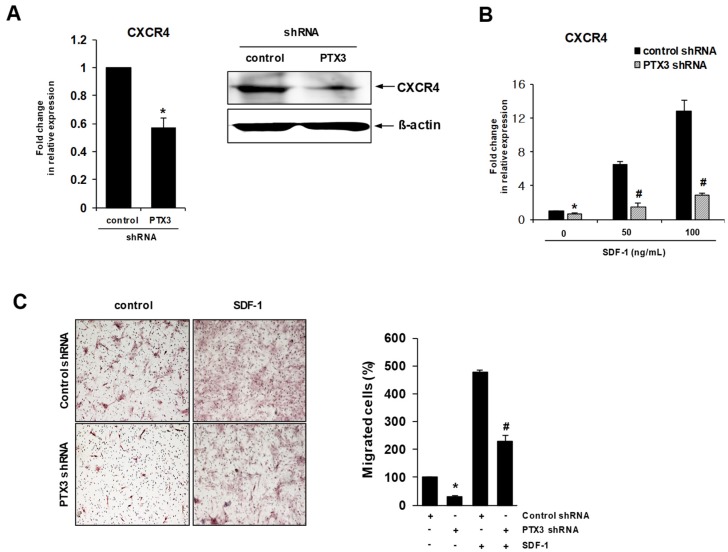
Effect of PTX3 knockdown on SDF-1α/CXCR4 axis in human dental pulp stem cells. (**A**) HDPSCs were stably transduced with lentiviruses encoding control shRNA or PTX3 shRNA. Total RNA was isolated and then analyzed using real-time PCR with primers specific for CXCR4. * *p* < 0.01 compared to control shRNA-transduced HDPSCs (left). Expression levels of CXCR4 protein in cell extracts were estimated by Western blotting (right). (**B**) Control shRNA-transduced HDPSCs and PTX3 shRNA-transduced HDPSCs were separately incubated with SDF-1 (50 ng/mL or 100 ng/mL) for 24 h. Total RNA was isolated and then analyzed by real-time PCR using CXCR4-specific primers. * *p* < 0.01 compared to untreated control shRNA-transduced HDPSCs; # *p* < 0.01 compared to SDF-1-treated control shRNA-transduced HDPSCs. (**C**) SDF-1 (50 ng/mL) was placed in the lower chambers of transwells and control shRNA- or PTX3 shRNA-transduced HDPCs were added to the upper chambers. The migrated cells were stained with hematoxylin-eosin and observed by a microscope at 100× magnification. The number of cells that migrated from the upper to the lower part of the transwells after 24 h was counted. * *p* < 0.01 compared to untreated control shRNA-transduced HDPSCs; # *p* < 0.01 compared to SDF-1-treated control shRNA-transduced HDPSCs.

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
