# Peer review of "Pentraxin-3 Modulates Osteogenic/Odontogenic Differentiation and Migration of Human Dental Pulp Stem Cells"

_ijms, 2019, doi:10.3390/ijms20225778_

Round 1

Reviewer 1 Report

In this paper the Authors characterized for the first time the biological role of PTX-3, a member of the Pentraxin superfamily, in human dental pulp stem cells. In particular, the Authors demonstrated

that PTX-3 silencing inhibits the migration and the differentiation of human dental pulp stem cells (HDPSC). These results were interesting, suggesting that PTX-3 could be involved in inflammatory-driven tissue homeostasis and may contribute to the therapeutic potential of HDPSC for regeneration and repair of dentin-pulp complex.

The study is on a timely subject in view of increasing interest about the dental pulp regenerative therapies (i.e. regenerative dentistry).

The techniques used (shRNA/lentiviral-based silencing, Real time PCR, Western blot, ELISA, transwell chemotactic and scratch wound migration assays) were appropriate and described with plenty details. This is a well-designed study with rigorous methods. The discussion is well-balanced and the statements are supported by the data.

Author Response

We corrected some spelling errors and newly added new sentences which provide a comprehensive understanding about migration of HDPSCs to the “Discussion” section and cited additional references in the revised manuscript.

Reviewer 2 Report

In the manuscript titled “Pentraxin-3 modulates osteogenic/odontogenic differentiation and migration of human dental pulp stem cells”, the authors evaluate function of Pentraxin-3 (PTX3) in human dental pulp stem cells (HDPSCs)through knockdown studies.

I would recommend the publication of the manuscript once the below comments are addressed.

Should "a-tubulin" in Figure 2A be "α-tubulin”?

Should "*" in Figure 2B legend be "#”? Please provide the right descriptions of "*" in the figures 2B legend.

The authors suggested that SDF-1 played a role in migration of HDPSCs. Where in normal tissue is SDF-1 expressed?

4.The argument of migration in “DISCUSSION” isn't enough. The authors should discuss the migration of HDPSCs based on SDF-1 expression in normal tissue.

I hope these comments will be helpful.

Author Response

Should "a-tubulin" in Figure 2A be "α-tubulin”?

: We correct “a-tubulin” to “α-tubulin” on Figure 2A as the reviewer commented.

Should "*" in Figure 2B legend be "#”? Please provide the right descriptions of "*" in the figures 2B legend.

: We add the descriptions of "#" to the “Figure legend” section and delete "*" in the figures 2B and legend section in the revised manuscript.

The authors suggested that SDF-1 played a role in migration of HDPSCs. Where in normal tissue is SDF-1 expressed?

   : Inflamed human dental pulp contained high levels of SDF-1 expression which positively detected in infiltrative inflammatory cells and endothelial cells of the microvessels near the carious regions (1). Other studies reported that SDF-1 expression were upregulated by odontogenic induction, inflammatory stimuli, and hypoxic signal in dental pulp cells (2,3). Meanwhile, there were no infiltrative inflammatory cells detectable in normal dental pulp and weak SDF-1 expression within the normal dental pulp (1,4). Li et al., mentioned SDF-1 is an important regulator of HDPSCs migration, though it remains to be elucidated whether it is actually secreted by stem cells as well as inflammatory cells (5).

* References

1) The expression of stromal cell-derived factor 1 (SDF-1) in inflamed human dental pulp. Jiang HW, Ling JQ, Gong QM. J Endod. 34(2008):1351-135.

2) Regulation of the stromal cell-derived factor-1alpha-CXCR4 axis in human dental pulp cells. Gong QM, Quan JJ, Jiang HW, Ling JQ. J Endod. 36(2010):1499-503.

3) The role of SDF-1 and CXCR4 on odontoblastic differentiation in human dental pulp cells. Kim DS, Kim YS, Bae WJ, Lee HJ, Chang SW, Kim WS, Kim EC. Int Endod J. 47(2014):534-41.

4) The expression and role of stromal cell-derived factor-1alpha-CXCR4 axis in human dental pulp. Jiang L, Zhu YQ, Du R, Gu YX, Xia L, Qin F, Ritchie HH. J Endod.34(2008): 939-44.

5) SDF-1/CXCR4 Axis Induces Human Dental Pulp Stem Cell Migration through FAK/PI3K/Akt and GSK3beta/beta-Catenin Pathways. Li M, Sun X, Ma L, Jin L, Zhang W, Xiao M, Yu Q. Sci Rep. 9(2017):40161.

  4.The argument of migration in “DISCUSSION” isn't enough. The authors should discuss the migration of HDPSCs based on SDF-1 expression in normal tissue.

: As the reviewer commented, we add new sentences which provide a comprehensive understanding about migration of HDPSCs based on SDF-1-CXCR4 axis to the “Discussion” section (page 9, line 196-216) and cite additional references (page 11, line 413-429) in the revised manuscript.

Round 2

Reviewer 2 Report

Thank you for adding discussion about the migration of HDPSCs based on SDF-1 expressionThe manuscript has been improved and is in a nice condition now.